# DL-BASED PREDICTION OF OPTIMAL ACTIONS OF HUMAN EXPERTS

## ABSTRACT

Expert systems have been developed to emulate human experts' decision-making. Once developed properly, expert systems can assist or substitute human experts, but they require overly expensive knowledge engineering/acquisition. Notably, deep learning (DL) can train highly efficient computer vision systems only from examples instead of relying on carefully selected feature sets by human experts. Thus, we hypothesize that DL can be used to build expert systems that can learn human experts' decision-making from examples only without relying on overly expensive knowledge engineering. To address this hypothesis, we train DL agents to predict optimal strategies (actions or action sequences) for the popular game 'Angry Birds', which requires complex problem-solving skills. In our experiments, after being trained with screenshots of different levels and pertinent 3-star guides, DL agents can predict strategies for unseen levels. This raises the possibility of building DL-based expert systems that do not require overly expensive knowledge engineering.

## 1 INTRODUCTION

To assist our decision-making, knowledge engineering has been developed to code human experts' knowledge into artificial expert systems, which have been successfully applied to multiple domains (Buchanan & Smith, 1988; Russell & Norvig, 1995; Kiritsis, 1995). For instance, MYCYN (Buchanan & Shortliffe, 1984) can provide medical advice on infectious diseases, and XCON (McDermott, 1980) can help users to find right computer components. However, knowledge engineering requires expensive and complex engineering, limiting the expert systems' utilities (Buchanan & Smith, 1988; Kiritsis, 1995). Notably, DL can enable artificial agents to automatically learn general rules essential for complex tasks from examples without human experts' instructions (LeCun et al., 2015; Hertz et al., 1991). This leads to the hypothesis that we can use DL to build expert systems without expensive knowledge engineering (Tan, 2017). Indeed, a line of study already explores DL agents' utility in diagnosing diseases based on medical images, for which traditional expert systems were used; see (Litjens et al., 2017) for a review.

However, it should be noted that most humans' decisions involve a sequence of actions, each of which has a subgoal. Thus, sequence learning can be a key component in building artificial intelligent agents that mimic human experts' decision-making (Sun & Giles, 2001; Boddy, 1996). Traditionally, symbolic planning is used to mimic humans' goal-directed high-level planning, which repeatedly creates sub-goals necessary to accomplish higher level goals until a subgoal can be accomplished in a single action, and deploying symbolic planning requires 'substantial prior knowledge' and highly computational complexity (Sun & Giles, 2001). That is, the cost of symbolic planning can be as high as that of other expert systems.

In this study, we explored if DL can be used to learn high-level planning essential for human experts' decision-making. We note that deep Q learning (Mnih et al., 2015) has demonstrated impressive achievements in learning optimal sequences of actions for games (Silver et al., 2016) and that recurrent neural networks have been used to model time series (Hertz et al., 1991). Our goal differs from these traditional DL approaches, in that we aim to model high-level planning more directly. To this end, we asked if deep neural networks (DNNs, agents trained by DL) can learn to predict strategies written by human experts for the popular game 'Angry Bird' (AB) (Contributors, 2021; Renz et al., 2019) from examples. To clear each stage, players must destroy all pigs in each stage.

As the pigs within the game are protected by piles of objects or stay inside towers, a single attack is often not enough to destroy all the pigs in each stage. Instead, players must sequentially organize subgoals to destroy all pigs efficiently, making AB an ideal testbed for high-level planning. Thus, we trained DNNs to predict human experts' AB game strategies available at AB wiki pages (Contributors, 2021); the AB strategies will be referred to as 3-star guides (3SG) hereafter.

In the experiments, we asked three questions. First, can DL learn the functional relationships between screenshots (still images) of the stages and 3SG (human experts' strategies)? Second, can an attention-based image captioning (ABIC) system, previously proposed (Xu et al., 2015), learn to create verbal strategies, which would sound reasonable to human players? Third, can ABIC predict a sequence of strategies (high-level planning) appropriate to clear a stage? Due to the lack of proper evaluation metric, we qualitatively evaluate the outcomes of the answers after training ABIC; the evaluation metrics of the language model is the topic of active research (Sellam et al., 2020; Papineni et al., 2001; Chen et al., 1998). Our empirical studies suggest that ABIC can learn to create verbal strategies after training and that it can predict a sequence of actions, not just a single action. They also support our hypothesis that DL can learn high-level planning from examples and that it can be used to build expert systems to assist human decision making without relying on overly expensive and complex knowledge engineering. Thus, our study indicates that DL can help us deploy expert systems into more domains.

## 2 DL agents to predict optimal strategies of Angry Birds (AB)

To clear each level of AB, players use birds to destroy the pigs that are scattered around and protected by towers or piles of wood, glass (or ice) and stone blocks. Each level provides a fixed sequence of birds and a unique challenging environment, and the players need to find an effective sequence of actions that can destroy all pigs in each level. The scores are estimated using various factors such as incurred damages and the number of birds used. When players obtain sufficiently high scores, they receive 3 stars. Any experienced game player (i.e., an expert) can devise good strategies (i.e., sequence of actions) to clear even the unseen levels, and conversely, the accuracy/efficiency of the plans can indicate the players' ability/efficiency. Therefore, we ask if DL could predict optimal actions (or action sequence) from given screenshots of levels. We use the 3SG available at the wiki page (Contributors, 2021) as optimal strategies for DL to predict them.

### 2.1 Functional correlations between screenshots and 3SG

In principle, we can consider an expert system as an unknown functional mapping function that can identify an optimal relationship between problems/situations (i.e., screenshots in this study) to decisions/solutions (i.e., 3SG in this study); see Fig. 1A. If the mapping does exist, it would be possible to build expert systems. Thus, we first investigate functional correlations between screenshots and 3SG. Since a multilayer perceptron (MLP) has been considered a universal problem solver (Hertz et al., 1991), we assume that a trained MLP can predict the embeddings of 3SG (word embeddings) from the embeddings of screenshots (image embeddings), if the functional mappings from the screenshots to 3SG exist.

Based on this assumption, we train MLPs, which contain 100 hidden neurons in single hidden layers, with 80% levels (out of 327) randomly selected and test them with the remaining (test) levels. The image embeddings are created using the ResNet18 (Safka, 2021), and the word embeddings are generated using Hugging Face's BERT model (Wolf et al., 2020). We first use image embeddings as inputs and word embeddings as outputs (Fig. 1B) and train MLP to minimize the mean squared error (MSE). A open-source ML library Pytorch (Paszke et al., 2017) is used to construct MLPs, and its built-in function (MSELoss) is used to estimate MSE. The word embeddings are projected to the PCA space, which is spanned by various numbers of principal axes, using a python package 'Scikit-learn' (Pedregosa et al., 2011). In the experiments, we repeat the same experiment 5 times with different random seeds and display the mean and standard deviations of them in Fig. 1C. As the number of principal axes increases, MLPs' errors are reduced, and the magnitudes of reduction grow bigger; see the blue line in Fig. 1C. Interestingly, when the word and image embeddings are used as inputs and outputs, MLP cannot predict the outputs anymore (the orange line in Fig. 1C). These results suggest that functional mappings from screenshots to 3SG exist, but not vice versa.

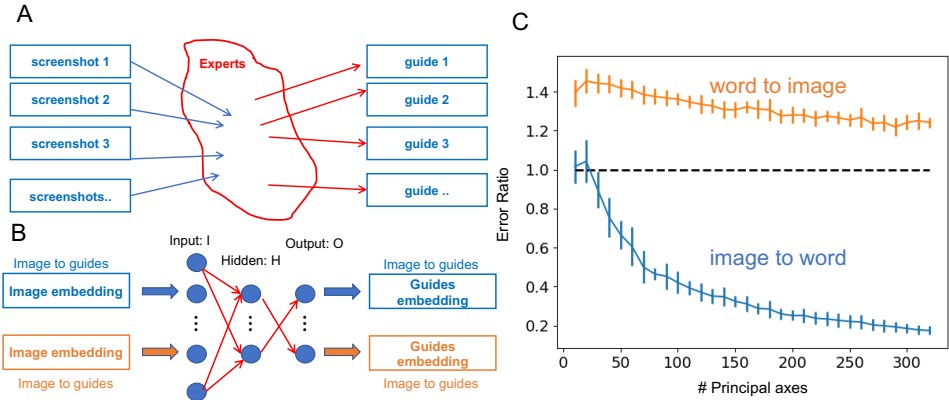

Figure 1: Functional mapping between image embeddings and word embeddings. (A), Schematics of expert system. (B), Diagram of MLP trained to detect functional mappings. In MLPs, single hidden layers contain 100 neurons. (C), MSE error ratio between before and after training. The MSE error estimated after training is normalized to that estimated before training. Blue and Orange represent error ratios for experiments 'img-to-word' and 'word-to-img', respectively.

## 2.2 EXPERT SYSTEMS OF AB

Encouraged by the results above, we ask if DL can directly predict human instructions from screenshots. Here, we train the attention-based image-captioning system (ABIC) (Xu et al., 2015) to predict 3SG from the screenshots. In the original form, ABIC uses convolutional networks as an encoder and recurrent networks as a decoder. The encoder extracts 14-by-14 annotation vectors, and the decoder uses annotation vectors to predict word sequences after training. They further proposed that the attention network, which determines the region of interest, can help the encoder-decoder system learn to produce captions. We refer to the original study for more details (Xu et al., 2015).

In this study, we adopted Pytorch implementation of ABIC from a public github repository (Vinodababu, 2021) and train it to predict 3SG from the screenshots using the default setting of the implementation. This adopted Pytorch implementation uses ResNet101 (He et al., 2015) as a encoder and the long short term memory (LSTM) (Hochreiter & Schmidhuber, 1997) as a decoder. Specifically, ResNet101 produces 2048 annotation vectors (14-by-14 pixels), the attentional network transforms annotation vectors using weighted sum, and LSTM selects the best word with previously selected word and weighted annotation vectors. In the experiments, we randomly select 70% of the levels as training examples, 15% as validation examples, and 15% as test examples. The number of total levels used here is 359. ResNet101 was adopted from TorchVision (Marcel & Rodriguez, 2010) and frozen during training, whereas both LSTM and the attentional network were trained.

### 2.2.1 ABIC CAN BE AN AB EXPERT

We note that ABIC was trained to produce a single sentence caption after being trained with multiple captions for a single image (Xu et al., 2015). To test the potential of AIBC as an AB expert, we mimic this baseline training of AIBC by ignoring the fact that 3SG describe the sequence of actions. Instead, we treat them as independently generated guides for the same stage and use them to train AIBC as an expert system of Angry Birds (EAB) to predict a single action. With this setting, we also effectively increase the number of training examples.

In AB and its wiki page (Contributors, 2021), levels are grouped into multiple stages, and the two consecutive numbers are used to refer to each level. For instance, level '12-03' refers to the third level in the 12th stage, and the same type of reference is used to clarify the levels below. In the

experiments, 253 levels are randomly chosen as training and 50 levels as test examples (i.e., levels). We edit the guides slightly for clarification and remove irrelevant descriptions. Many instructions explain the bird types, while most screenshots do not include the birds. Thus, we augment the screenshots by inserting the circles that represent the first birds in the levels via colors in the top-left corners (Fig. 2).

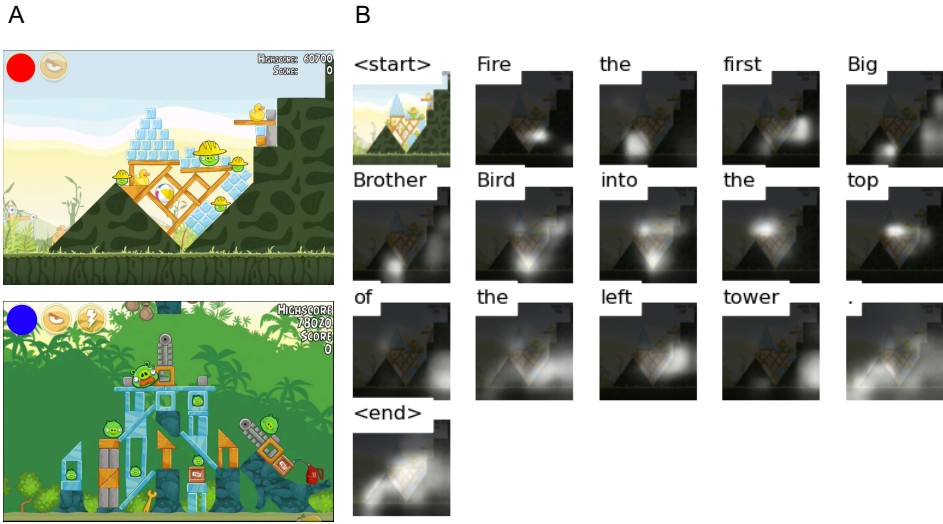

Figure 2: Prediction of EAB on the single action. (A), Examples of screenshots augmented with bird symbols; see top-left corers. (B), Example of outputs of ABIC trained as EAB.

With the edited guides and augmented screenshots, we train EAB and evaluate its predicted actions for test (unseen) screenshots. Fig. 2B shows an example of EAB outputs. As shown in the figure, the model utilizes attention to create a strategy (sentence). Table 1 shows some examples of EAB's predicted actions/strategies for the unseen test levels (i.e., screenshots). Here, we make a few observations. First, although the guides describe the sequence of actions, most EAB outputs appear to describe only the first action. Second, EAB selects the correct birds to create predictions on strategies for 10 test levels. Third, a few of the predicted strategies are consistent with the guides. Fourth, the 3SG have diverse writing styles, making the quantitative evaluation of predicted strategies extremely difficult.

These results raise the possibility that DL can learn to predict an optimal action from examples only. Due to EAB's low precision, we tested if EAB's performance can be further improved..

### 2.2.2 EAB BECOMES MORE ACCURATE WITH BETTER OBJECT DETECTION UNITS

In the experiments mentioned above, the screenshots are fed to the image encoder (ResNet101) trained with ImageNet (Xu et al., 2015; Vinodababu, 2021). However, it should be noted that the visual objects used in AB are abstract and substantially different from real world objects in the ImageNet, which could reduce the quality/precision of visual embeddings of the ResNet101 and consequently the precision of EAB. Thus, we ask if better image embeddings can improve EAB's accuracy. In doing so, we build customized detectors of the objects in the game, which use color templates of pigs, glass/ice, wood and stone blocks. Specifically, we use OpenCV's 'inRange' function (Itseez, 2015; Its, 2014) to find candidate areas for the objects and scipy.ndimage's 'find_object' (Virtanen et al., 2020) to test if the areas contain actual objects. For more accurate detection, we manually adjust the templates of individual stages and use a few simple error correction rules. After detecting the objects, we use them to synthesize functional screenshots of the stages (Fig. 3B).

After training EAB with synthesized (reconstructed) screenshots and 3SG of 249 training levels, we qualitatively evaluate EAB's predictions for 58 test levels. We first select the predictions that articulate the correct birds and categorize them into 3 different categories, not-related strategy (NRS), probable strategy (PS), consistent strategy (CS). NRS indicates strategies not relevant to the levels,

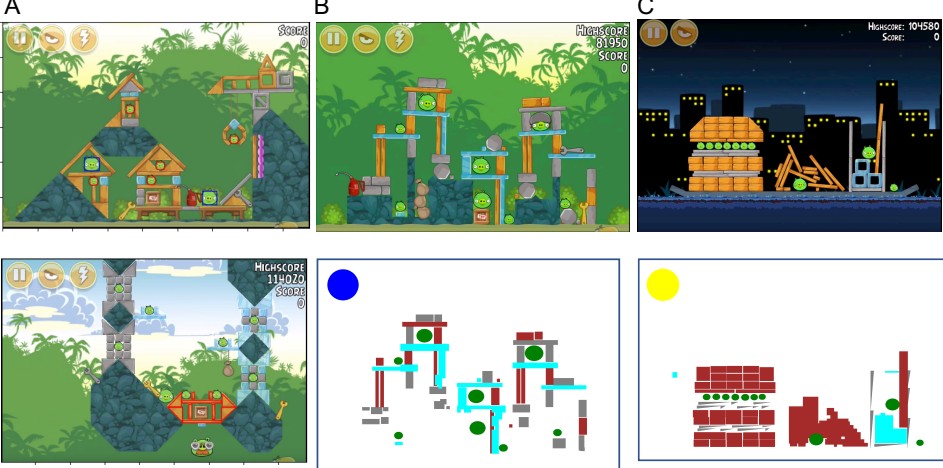

Figure 3: Detection of game objects. (A), Examples of detected pigs (top-row) and wood blocks (bottom-row). Our detectors can distinguish the size of pigs. Blue and red boundary boxes refer to mid and small pigs. (B) and (C), Examples of raw screenshots (top-row) and reconstructed images (bottom-row).

PS indicates strategies reasonable (not necessarily good) for the levels, and CS indicates strategies consistent with 3SG. We note that some CS do not specify targets and thus code them as CS-ND (consistent strategies without detailed targets).

We found that 41 levels have the correct birds. 6 levels are evaluated as CS; 16 levels as PS, 5 levels as CS-ND amd 14 levels as NRS. Table 2 shows the examples of all 4 categories.

### 2.2.3 EAB CAN PROPOSE MULTIPLE ACTIONS

In AB, players can obtain optimal results by strategically organizing sequential actions. Thus, as the final test, we ask if EAB can learn to create a sequence of actions. Specifically, we remove the 'end' token from 3SG to build embeddings of action sequences. After training EAB with 253 training levels, we evaluate the predicted sequence for 50 test levels and make a few observations (Table 3). First, EAB predicts multiple actions for most levels—1 action in 10 levels, 36 actions in and 3 actions for 4 levels. Second, EAB can choose a sequence of birds properly in some levels. Specifically, it selects two first birds correctly for 13 levels, 3 first birds correctly for 2 levels, and 2 birds for 5 levels. Third, the accuracy of the predicted sequence of actions is low, but EAB still predicts action sequences consistent with 3SG for 3 levels. EAB also suggests probable action sequences for 6 levels. For 5 levels, one of the predicted actions includes some minor errors. For instance, EAB proposes to aim 'wooden triangle' for level '4-04', but it does not exist in this level. We also found some predicted actions do not specify the targets for 3 levels. We categorize them into CS, PS, PS without targets (PS-ND) PS with minor errors (CS-me). Table 3 list the examples of CS, PS, PS-ND, PS-me and NRS.

## 3 DISCUSSION

In this study, we used the popular game Angry Bird to study the plausibility of DL-based expert systems. Specifically, we trained an off-shelf model (ABIC) to predict human players' strategies (i.e., experts). Even though the strategies are written by many players, and thus they are heterogeneous in styles and vocabularies, ABIC was able to predict optimal strategies for given screenshots of stages. This result raises the possibility that DL can be used to build expert systems without overly expensive knowledge engineering and make the expert systems more affordable and deployable to more domains.

Further, our results suggest that DL can encode human experts' knowledge/skills from their verbal explanations, leading us to speculate that DL can be used to quantify or better understand human behaviors. Below we briefly discuss two possible scenarios.

## 3.1 IMPLICATION FOR INTENTION DETECTION

By predicting others' intentions, we can make decisions and take actions. When we detect dangerous behavioral cues from someone approaching us, we natively predict hostility and coordinate our actions accordingly. If an advanced surveillance camera can detect hostile intentions in public, they may proactively alert authorities to prevent criminal activities or neutralize threats. However, the behavioral cues that we rely on remain poorly understood, making the explicit design of intention detectors extremely challenging. Based on our experiments suggesting that DL can learn to predict humans' decision-making process from examples, we argue that DL can be used to build intention detectors such as advanced surveillance systems.

## 3.2 IMPLICATION FOR TRAINING

To become experts, students need extensive training and experience. However, teaching requires experts to translate their experience/knowledge into standard and quantifiable languages. This remains a challenging process in many domains. Our results (Fig. 1C) suggest that we could build functional mappings between experts' behaviors and tasks. If we could build DL agents to capture functional relationships between expert's decisions and tasks' components, we could use them to translate experts' knowledge into quantifiable forms, which can facilitate students' learning. For instance, we can build an expert system that can faithfully predict veteran pilots' decisions and use it to analyze the influence of aircrafts' sensors on their decisions. The discovered correlations between aircrafts' sensors and pilots' decisions will prepare student pilots more effectively.

## 3.3 LIMITATION AND FUTURE DIRECTION

Our EAB is trained to predict the optimal strategy (either a single action or a sequence of actions) for the level, but multiple strategies can be used to accomplish the same goal, and the best or optimal choice varies depending on contexts. Therefore, real world expert systems must learn various types of strategies to choose the optimal strategy. To this end, we need to build multiple agents specializing in individual strategies and develop dynamic evaluation systems that can flexibly choose the best agent from the available pool. In the future, we plan to seek such multi-agent expert systems.

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

Table 1: This table lists predicted actions with the correct bird and bad predictions. We also list the first action in 3SG for comparison.

| Predictions with the correct bird | | |
|---|---|---|
| levels | Predicted action | First action in 3SG |
| 8-07 | Fire the first Black Bird into the left side of the main structure. | Fire the Black bird into the hollow stone square of the first tower, pushing both stone towers to the right and popping all pigs on the right side. |
| 12-05 | Fire the first Big Brother Bird to the left side of the second structure. | Fire Big Brother through the top half of the first wooden wheel. |
| 12-09 | Fire the first Red bird into the top of the left tower, causing the structure to fail to the left. | Fire Big Brother through the horizontal wooden plank above the first pig. |
| 15-02 | Fire the first Black Bird into the horizontal stone plank on the right side. | Shoot the Black Bird in the middle of the level to activate the TNT. |
| 4-04 | First, fire the Yellow Bird so he <unk>hits the second stone plank | First, send the Yellow Bird through the second angled wooden plank. |
| 5-11 | Fire the first Yellow Bird into the first horizontal stone plank. | First, fire the Yellow Bird so he breaks the horizontal wood plank supporting the stone plank |
| 2-06 | Lob the first Black bird into the hollow stone square of the right side of the level | Shoot the first Black Bird so that it lands directly to the right of the highest pig. |
| 10-13 | Fire the first Big Brother into the bottom-left of the main structure. | Fire the first Big Brother into the middle of the all-stone tower (which has the Medium Pig and a round wooden ball under the tower). |
| 11-11 | Fire the first Big Brother through the horizontal wooden plank between the first <unk>. | Fire the first Big Brother into the horizontal stone planks in front of the slingshot platform. |
| 9-08 | Fire the first Big Brother into the top of the left tower | First, fire the first Big Brother Bird head-on into the top-left pig. |
| Bad examples | | |
| | Prediction | First action in 3SG |
| 13-05 | Fire the first Red bird into the leftmost structure, spinning it back through the the TNT in the right. | Lob one or both Brothers high in the air, crashing through the middle of the structure. |
| 15-10 | Fire the first Red bird into the leftmost structure, spinning it back through the the TNT in the right. | Fire the first Yellow bird through the wooden block (not the hollow square) on the left side. |
| 1-03 | The first Red Bird at the <unk>of the <unk>pig. | Fire the first Red Bird into the wooden sphere (ball) on top of the hill. |

Table 2: Below we list all CS and CS-ND and some examples of PS and NRS. EV and LV refers to evaluation and level.

| EV | Predicted action | First action in guide | LV |
|---|---|---|---|
| CS | Fire the Boomerang bird into the main structure, spinning it back into the TNT. | Fling the Boomerang bird over the main structure, spinning it back into the TNT. | 22-02 |
| CS | Fire the first Red bird into the horizontal stone block on the first structure. | Fire the Red bird into the top of the left structure, pushing its roof to the right. | 12-03 |
| CS | Launch the first Blue bird into the middle of the ice tower. | Shoot the Blue Bird at the glass house to the right. | 15-04 |
| CS | Shoot the White Bird at the top of the structure, bombing as late as possible. | Shoot the first White Bird's egg at the stone blocks to the right of the TNT. | 2-20 |
| CS | Launch the first Big Brother into the bottom of the ice tower. | Fire the Big Brother Birds straight ahead, aiming for the TNT beneath the boat. | 10-14 |
| CS | Fire the first Big Brother into the bottom of the first tower, taking it to the right. | Fire the Red bird through the first horizontal wooden plank, causing the first column to collapse. | 12-08 |
| CS-ND | Shoot the first Black Bird through the gap between the two towers. | Loft the Black bird to land dead-center on the bridge. | 12-15 |
| CS-ND | Fire the first Yellow Bird into the left side of the first structure. | Shoot the Yellow Bird at the lone wood block surrounded by stone. | 2-13 |
| CS-ND | Shoot the Red Bird at the middle of the structure. | Launch the Big Brother Bird through the first pig and into the first horizontal stone block and the entire structure will crumble. | 9-04 |
| CS-ND | Use the Boomerang bird over the main structure, spinning it back through the stone blocks on the right side. | Launch a Boomerang Bird. with perfect timing, boomerang it into a stone brick on the upper "cannon" to ram the stone into the TNT, killing the Medium pig. | 6-06 |
| PS | Fire the first Big Brother through the horizontal wooden block anove the first structure | Launch Big Brother Bird through the small ice square in the top-left of the structure. | 9-05 |
| PS | Fire the Yellow bird horizontally into the gap in the middle of the level | Fire the first Yellow bird into the top-left TNT. | 17-14 |
| PS | Fire the first Big Brother through the top of the small wooden squares on the bottom-of the structure. | Use the Red bird to topple the stones atop the tower, clearing the pigs on the right side of the level. | 22-08 |
| PS | Loft the Yellow Bird to break the TNT In the <unk>of the <unk>. | Loft a Yellow bird and accelerate through the middle structure. | 12-04 |
| PS | Fire the first Big Brother into the bottom of the first tower. | fire Big Brother through the horizontal wooden plank above the first pig. | 12-09 |
| NRS | Fire the first Big Brother through the top of the small wooden squares on the bottom-of the structure. | Use the Red bird to topple the stones atop the tower, clearing the pigs on the right side of the level. | 22-08 |
| NRS | Fire the first Black bird into the top of the pig, causing the TNT to the right | Fire the Black bird downward into the pile of square stones, launching them rightward and popping all pigs on the left side of the level. | 13-02 |
| NRS | Shoot the Yellow Bird so that it so that it hits the TNT in the middle of the level. | First, shoot the Yellow Bird through the second level of the triangle block, eliminating the first pig. | 4-06 |
| NRS | Use the Boomerang bird to blast the wood blocks <unk>. | Launch the first Hal upwards. With precise timing, activate the boomerang mode to hit the middle layer of wood. | 6-05 |

Table 3: We list examples of prediction of sequence of actions. All 3 CS are displayed.

| EV | Predicted action | Action sequence in guide | LV |
|---|---|---|---|
| CS | Fire the first Yellow Bird into the middle of the right tower. Use the next Yellow bird through the wooden plank in the middle of the structure. Use the next Yellow Bird to take the remaining pigs on the left side. | Fire the first Yellow bird through the wooden block (not the hollow square) on the left side. Launch the next Yellow bird through the wood in the middle of the level. If pigs remain, you have a Yellow bird and Big Brother to clean up. | 15-10 |
| CS | Fire the first Black bird into the top of the structure, causing the main tower to fall. Then use the next Black bird, landing it to clear the remaining pigs. | Fling the bomb bird directly into the left helmet pig, and trigger upon impact. Then drop the second bomb bird in the gap between the two lower structures. | 23-03 |
| CS | First, use the White Bird to bomb the middle of the structure, bombing as late as possible. Then use a Yellow Bird through the wooden blocks in the middle of the level. | Shoot the first White Bird's egg at the stone blocks to the right of the TNT. | 2-20 |
| PS | First, fire the Yellow Bird to the left of the left structure. Then, shoot the Yellow Bird through the middle of the structure, popping all three pigs. | Shoot the first Yellow Bird next to the Small Pig that stands next to the TNT at the top. Shoot the second Yellow Bird to the remaining pigs to clear the level. | 14-01 |
| PS | Fire the first Big Brother through the highest wooden block on the right side of the tower to fall. | Fire Big Brother straight ahead. Use the Boomerang bird to attack the right side of the level from behind, then fire another Brother forward to pop all remaining pigs. | 13-15 |
| PS | Lob the first Black bird into the gap between the first two towers. Fire the next Black bird into the left side of the structure, causing it to topple. Fire the next Black bird into the left side of the structure, causing it to topple. | Fire the first Black bird into the bottom-left support. Arc the next Black bird into the small pig in the top-right and the explosion should topple everything that remains. | 16-05 |
| PS-me | Fire the first Red Bird so he lands in the middle of the structure, causing it to topple. Fire the next Black bird through the stone triangle of the structure to finish. | Fire Big Brother through the horizontal wooden plank above the first pig. Fire the first Black bird along a similar trajectory, blasting the left side of the second-tallest tower. Launch the last Black bird a little lower, under the overhanging plank and into the base of the tall tower. | 12-09 |
| PS-me | First, fire the Yellow Bird at the left hollow wooden squares. Then shoot the Yellow Bird at the top hollow stone squares. | Shoot the first blue bird into the air, then aim it so it hits the wood between the first couple of stone blocks and the boulder. | 1-20 |
| PS-ND | Send the White bird towards the top structure. Then use the White bird to the right tower. | Use the White Bird's ability to reach the stone blocks hanging by string. Shoot the next White Bird to the wood "house" closest to the slingshot. Before you get hit, use the egg bomb. Shoot the Yellow Bird at the TNT. Shoot the Red Bird then to the last pig in the stone house. | 15-05 |
| NRS | Shoot the first Black Bird so that it hits the first pig. Shoot the next Black bird through the gap, spinning it back through the small wooden block above the next two pigs. | Shoot the first Black Bird so that it lands directly to the right of the highest pig. | 2-06 |
| NRS | Fire the Blue bird to break the two wooden blocks in the bottom-left of the structure. Fire the Blue bird through the left side of the structure, causing the TNT within. | Fling the Red bird into the ice leg of the rightmost tower. There can be an extreme amount of secondary destruction. In some cases, only one or two pigs remained. | 13-04 |

John Hertz, Anders Krogh, and Richard Palmer. *Introduction To The Theory Of Neural Computation*. SANTA FE INSTITUTE IIN THE SCIENCES OF COMPLEXITY. CRC Press, 1991. ISBN ISBN-13: 978-0367091361.

Sepp Hochreiter and Jürgen Schmidhuber. Long Short-Term Memory. *Neural Computation*, 9 (8):1735–1780, 11 1997. ISSN 0899-7667. doi: 10.1162/neco.1997.9.8.1735. URL https://doi.org/10.1162/neco.1997.9.8.1735.

*The OpenCV Reference Manual*. Itseez, 2.4.9.0 edition, April 2014.

Itseez. Open source computer vision library. https://github.com/itseez/opencv, 2015.

Dimitris Kiritsis. A review of knowledge-based expert systems for process planning. Methods and problems. *The International Journal of Advanced Manufacturing Technology*, 10(4):240–262, July 1995. ISSN 0268-3768, 1433-3015. doi: 10.1007/BF01186876. URL http://link.springer.com/10.1007/BF01186876.

Yann LeCun, Yoshua Bengio, and Geoffrey Hinton. Deep learning. *Nature*, 521(7553):436–444, May 2015. ISSN 0028-0836, 1476-4687. doi: 10.1038/nature14539. URL http://www.nature.com/articles/nature14539.

Geert Litjens, Thijs Kooi, Babak Ehteshami Bejnordi, Arnaud Arindra Adiyoso Setio, Francesco Ciompi, Mohsen Ghafoorian, Jeroen A.W.M. van der Laak, Bram van Ginneken, and Clara I. Sánchez. A survey on deep learning in medical image analysis. *Medical Image Analysis*, 42: 60–88, 2017. ISSN 1361-8415. doi: https://doi.org/10.1016/j.media.2017.07.005. URL https://www.sciencedirect.com/science/article/pii/S1361841517301135.

S. Marcel and Y. Rodriguez. Torchvision the machine-vision package of torch. *Proceedings of the 18th ACM international conference on Multimedia*, 2010.

John McDermott. R1: An expert in the computer systems domain. In *First AAAI Conference on Artificial Intelligence*, pp. 269–271. AAAI, 1980.

Volodymyr Mnih, Koray Kavukcuoglu, David Silver, Andrei A. Rusu, Joel Veness, Marc G. Bellemare, Alex Graves, Martin Riedmiller, Andreas K. Fidjeland, Georg Ostrovski, Stig Petersen, Charles Beattie, Amir Sadik, Ioannis Antonoglou, Helen King, Dharshan Kumaran, Daan Wierstra, Shane Legg, and Demis Hassabis. Human-level control through deep reinforcement learning. *Nature*, 518(7540):529–533, February 2015. ISSN 1476-4687. doi: 10.1038/nature14236. URL https://doi.org/10.1038/nature14236.

Kishore Papineni, Salim Roukos, Todd Ward, and Wei-Jing Zhu. BLEU: a method for automatic evaluation of machine translation. In *Proceedings of the 40th Annual Meeting on Association for Computational Linguistics - ACL '02*, pp. 311, Philadelphia, Pennsylvania, 2001. Association for Computational Linguistics. doi: 10.3115/1073083.1073135. URL http://portal.acm.org/citation.cfm?doid=1073083.1073135.

Adam Paszke, Sam Gross, Soumith Chintala, Gregory Chanan, Edward Yang, Zachary DeVito, Zeming Lin, Alban Desmaison, Luca Antiga, and Adam Lerer. Automatic differentiation in PyTorch. In *NIPS 2017 Workshop Autodiff*, 2017.

F. Pedregosa, G. Varoquaux, A. Gramfort, V. Michel, B. Thirion, O. Grisel, M. Blondel, P. Prettenhofer, R. Weiss, V. Dubourg, J. Vanderplas, A. Passos, D. Cournapeau, M. Brucher, M. Perrot, and E. Duchesnay. Scikit-learn: Machine learning in Python. *Journal of Machine Learning Research*, 12:2825–2830, 2011.

Jochen Renz, XiaoYu Ge, Matthew Stephenson, and Peng Zhang. AI meets Angry Birds. *Nature Machine Intelligence*, 1(7):328–328, July 2019. ISSN 2522-5839. doi: 10.1038/s42256-019-0072-x. URL http://www.nature.com/articles/s42256-019-0072-x.

Stuart J. Russell and Peter Norvig. *Artificial intelligence: a modern approach*. Prentice Hall series in artificial intelligence. Prentice Hall, Englewood Cliffs, N.J, 1995. ISBN 978-0-13-103805-9.

Christian Safka Safka. img2vec, 2021. URL https://github.com/christiansafka/img2vec.

Thibault Sellam, Dipanjan Das, and Ankur P. Parikh. BLEURT: Learning Robust Metrics for Text Generation. *arXiv:2004.04696 [cs]*, May 2020. URL http://arxiv.org/abs/2004.04696. arXiv: 2004.04696.

David Silver, Aja Huang, Chris J. Maddison, Arthur Guez, Laurent Sifre, George van den Driessche, Julian Schrittwieser, Ioannis Antonoglou, Veda Panneershelvam, Marc Lanctot, Sander Dieleman, Dominik Grewe, John Nham, Nal Kalchbrenner, Ilya Sutskever, Timothy Lillicrap, Madeleine Leach, Koray Kavukcuoglu, Thore Graepel, and Demis Hassabis. Mastering the game of Go with deep neural networks and tree search. *Nature*, 529(7587):484–489, January 2016. ISSN 1476-4687. doi: 10.1038/nature16961. URL https://doi.org/10.1038/nature16961.

R. Sun and C.L. Giles. Sequence learning: from recognition and prediction to sequential decision making. *IEEE Intelligent Systems*, 16(4):67–70, 2001. doi: 10.1109/MIS.2001.1463065.

Haocheng Tan. A brief history and technical review of the expert system research. *IOP Conference Series: Materials Science and Engineering*, 242:012111, September 2017. doi: 10.1088/1757-899x/242/1/012111. URL https://doi.org/10.1088/1757-899x/242/1/012111. Publisher: IOP Publishing.

Sagar Vinodababu. a-PyTorch-Tutorial-to-Image-Captioning, 2021. URL https://github.com/sgrvinod/a-PyTorch-Tutorial-to-Image-Captioning.

Pauli Virtanen, Ralf Gommers, Travis E. Oliphant, Matt Haberland, Tyler Reddy, David Cournapeau, Evgeni Burovski, Pearu Peterson, Warren Weckesser, Jonathan Bright, Stefan J. van der Walt, Matthew Brett, Joshua Wilson, K. Jarrod Millman, Nikolay Mayorov, Andrew R. J. Nelson, Eric Jones, Robert Kern, Eric Larson, CJ Carey, lhan Polat, Yu Feng, Eric W. Moore, Jake Vand erPlas, Denis Laxalde, Josef Perktold, Robert Cimrman, Ian Henriksen, E. A. Quintero, Charles R Harris, Anne M. Archibald, Antonio H. Ribeiro, Fabian Pedregosa, Paul van Mulbregt, and SciPy 1. 0 Contributors. Scipy 1.0: Fundamental algorithms for scientific computing in python. *Nature Methods*, 2020.

Thomas Wolf, Lysandre Debut, Victor Sanh, Julien Chaumond, Clement Delangue, Anthony Moi, Pierric Cistac, Tim Rault, Rémi Louf, Morgan Funtowicz, Joe Davison, Sam Shleifer, Patrick von Platen, Clara Ma, Yacine Jernite, Julien Plu, Canwen Xu, Teven Le Scao, Sylvain Gugger, Mariama Drame, Quentin Lhoest, and Alexander M. Rush. Transformers: State-of-the-art natural language processing. In *Proceedings of the 2020 Conference on Empirical Methods in Natural Language Processing: System Demonstrations*, pp. 38–45, Online, October 2020. Association for Computational Linguistics. URL https://www.aclweb.org/anthology/2020.emnlp-demos.6.

Kelvin Xu, Jimmy Ba, Ryan Kiros, Kyunghyun Cho, Aaron Courville, Ruslan Salakhudinov, Rich Zemel, and Yoshua Bengio. Show, attend and tell: Neural image caption generation with visual attention. In Francis Bach and David Blei (eds.), *Proceedings of the 32nd International Conference on Machine Learning*, volume 37 of *Proceedings of Machine Learning Research*, pp. 2048–2057, Lille, France, 07–09 Jul 2015. PMLR. URL https://proceedings.mlr.press/v37/xuc15.html.

