# OpenReview forum: "DL-based prediction of optimal actions of human experts"
_ICLR.cc/2022/Conference — ICLR 2022 Submitted_

### Official Review · Reviewer_iZWN · 2021-10-29

**Correctness:** 1
**Technical Novelty And Significance:** 2
**Empirical Novelty And Significance:** 2
**Recommendation:** 5
**Confidence:** 4

**Main Review:**

The paper studies a very interesting problem and has detailed discussions about the approach it pursues, but I have several concerns listed below:

[Statements not supported]
- The contribution of the paper and the difference from the previous work is not really well described. In the third paragraph of the introduction for example, it is written: "Our goal differs from these traditional DL approaches, in that we aim to model high-level planning more directly". This sounds very vague to me. It was better if you had described in a more scientific way what you mean by "more directly".

- In the end of page 2 you mention that from the word embeddings it is not possible to get to the image embeddings without investigating any further why that could be the case. Did you try any other architecture for this task and no other model was able to learn this mapping or you made this conclusion based on the result of the same model. In general, I find the last paragraph of page 2 a bit unorganized.

- There is no discussion on the relationship between the number of principal components chosen in PCA and the Error. I see that you conclude that more principal components result in lower error (Figure 1) but I see no more discussion on why that is the case.

[Lack of scientific writing/mathematical modeling]
- In the second paragraph of 2.1 where you start to say how you train your MLP to learn the embeddings it is better if you are more precise than saying "we train the MLP to minimize the mean squared loss". I assume what you mean is the MSE between the predicted word embedding and the ground truth embedding but it was better if you had defined the embedding vectors previously and had written the exact objective for training the MLP and then the MSE loss. I understand that you try to explain the training procedure verbally but in some cases it is much more elegant if you use mathematical notation. For example, in each of the three questions that you try to answer, you could first mathematically model the objective and then say how you optimize it.

[Not enough details regarding implementation]
- What is the architecture of the MLP you use?  What are the size of the image and word embeddings?

[Questions]
- What do you mean by error ratio in Figure 1. I cannot find the definition.
- You mention in the Discussion that ABIC is able to predict optimal strategies given the screenshots but you never defined what you mean by optimal and you never reported the accuracy that you achieve. How did you conclude that it is the optimal strategy?
- In none of the three tasks the performance of the machine in terms of the metric that you consider (I assume MSE) is not reported for either of the train/val/test sets. It is not possible to comment on the performance of the model when this is the case.

[Typos]
- ln the end of line 4 in 2.2.3
- I find the discussions in 3.1 and 3.2 a bit out of context and not related to the paper.

[Eplainability]
- I was expecting more examples of the outputs of the model. One of the goals of the paper is to study whether the instructions output by the model sounds reasonable to a human expert but this is not well discussed in the paper.


**Summary Of The Paper:**

The paper aims to study the training of Deep Learning (DL) models from examples and to this end trains a DL agent to predict actions given screenshots of the game "Angry Birds" and observe that the trained model is able to predict actions for test samples and concludes that possibly DL expert systems can learn only by examples. The contributions are three-fold:

1- First, they train an MLP model to learn the mapping from screenshots of the game to the word embeddings which correspond to the guide provided by a player on that stage of the game.

2-Trains an Attention Based Image Captioning System (ABIC) to predict human strategies from the screenshots of the game.

3- Trains ABIC to learn the sequence of strategies necessary to pass a stage.

**Summary Of The Review:**

The paper studies a very interesting problem but in my opinion, it lacks scientific writing and reasoning in some instances. It can be further improved if it is written in a more organized way where the objectives are defined and the error and the structure of the model are clearly reported. Please refer to my main review where I have addressed these concerns one by one.

---

> ### Author Response · Authors · 2021-11-22
> **response to reviewer 4**
>
> We would like to thank the reviewer's efforts and feedback.
>
> 1. Reinforcement learning (RL) has been used to train DL models to play AB (or other games). Although RL is a model of human learning, it remains unclear if RL can fully capture humans’ learning capability. More specifically, we organized a sequence of sub-goals to complete the tasks, which cannot be fully described by the rewards maximization. Thus, using AB as a testbed, we aimed to test if DL models can mimic our ability to plan.
>
> 2. For the experiment discussed in Section 2.1, we only used MLP because it is the simplest neural network model but powerful enough to learn a wide range of applications. Instead of testing multiple models, we focused on testing neural networks’ capacity to learn mappings from images to word embeddings. Our results can be summarized as follows. First, the errors between predicted word embeddings and ground truth embeddings, which are measured with MSE loss, decrease, as more principal axes are used. When more principal axes are used to represent images, image embeddings become more accurate. Thus, we can expect that a good learner (i.e., artificial agents) can predict strategies better, when more principal axes are used, which is supported by our experiment. Conversely, this result supports that neural networks can learn to predict human players’ strategy from screenshots. Second, the errors on word-to-image embeddings do not decrease. This experiment can be considered as a control experiment and indicates that the inference of images from word embeddings is quite difficult. Conversely, this result suggests that predicting human players’ strategies from screenshots would be within the reach of DL’s learning capability.
>
> 3. The errors shown in Fig. 1C are normalized to the error measured with the smallest number of principal axes. Additionally, to compare the image-to-word and word-to-image mappings, the errors of both mappings are normalized to the same reference point.
>
> 4. In the experiments, we implicitly assume that the 3-star strategies are approximately ‘optimal’ and qualitatively evaluate if the predicted strategies are equivalent to these quasi-optimal strategies. We use this qualitative measure due to the lack of universal evaluation metrics of language models.
>
> 5. Regarding the experimental protocol, 1) we trained MLP with the fixed epochs and 2) used the pre-defined experimental protocol, which is included in the ABIC implementation, to train EAB. We agree with the reviewer that they should be stated more clearly with mathematical definitions, which will be addressed in our future manuscript.
>
> 6. We agree that more examples of predictions would be beneficial. Since we spent 3 full pages to show representative examples in the tables, we decided not to show more examples. Currently, we are considering better ways to report our results with quantitative metrics.
>
> 7. Lastly, we would like to thank the reviewer for detailed feedback and suggestions.

---

### Official Review · Reviewer_6Hog · 2021-11-02

**Correctness:** 1
**Technical Novelty And Significance:** 2
**Empirical Novelty And Significance:** 2
**Recommendation:** 3
**Confidence:** 4

**Main Review:**

The paper tackles an interesting problem, which might be situated in or in between learning from demonstrations [0] or imitation learning variants and learning from activity descriptions [1].

The paper, however, never clearly and formally defines the learning objective and/or situates the research into such established frameworks/learning protocols. The paper only shortly mentions the standard RL setting and symbolic planning, but a clear relationship to prior, relevant works is not available.

The empirical results are only shown in interpretable numbers for an initial experiment for testing the feasibility of prediction word embeddings from image embeddings. There are no clear results for the angry birds experiments for predicting instructions.

[0] Hester, T., Vecerik, M., Pietquin, O., Lanctot, M., Schaul, T., Piot, B., Horgan, D., Quan, J., Sendonaris, A., Osband, I. and Dulac-Arnold, G., 2018, April. Deep q-learning from demonstrations. In Thirty-second AAAI conference on artificial intelligence.
[1] Nguyen, K., Misra, D., Schapire, R., Dudík, M. and Shafto, P., 2021. Interactive Learning from Activity Description. arXiv preprint arXiv:2102.07024.

**Summary Of The Paper:**

The paper proposes an approach for building an expert system for sequential decision making problems from image-represented states and instruction-represented actions.  The authors reuse pretrained image- and word embeddings to train a MLP for predicting instructions from game frames. The paper specifically addresses the game angry birds and reports good results for mapping image embeddings to word embeddings.

**Summary Of The Review:**

Paper does not clearly formalize the problem, misses to draw concise relationships to established, relevant prior works, and does not report the empirical results in an interpretable form.

---

> ### Author Response · Authors · 2021-11-22
> **response to reviewer 3**
>
> We appreciate the reviewers’ feedback and suggested references. We did not aim to train DL models to play AB properly. Instead, we focused on testing DL models’ ability to predict human players’ strategies from screenshots. We implicitly assumed that if DL models could learn our high-level planning, they could produce verbal statements qualitatively equivalent to human players’ strategies. We believe that our results raise the possibility that even off the shelf DL models are powerful enough to mimic humans’ high-level planning.
>
> However, we agree with the reviewer that our study would greatly benefit by discussing its relationships with the earlier studies (focusing on training DL models to become the best game players) and presenting more quantitative metrics. We plan to incorporate the reviewer’s suggestions into our future works.

---

### Official Review · Reviewer_QBNS · 2021-11-02

**Correctness:** 3
**Technical Novelty And Significance:** 1
**Empirical Novelty And Significance:** 1
**Recommendation:** 1
**Confidence:** 4

**Main Review:**

This paper is clearly not ready for publication. It has a number of extremely large issues. The only strength of the paper is that the task of learning human guides from visual inputs is potentially interesting. However, otherwise the execution of this paper is very flawed.
- Why just Angry Birds? The evaluation should be much more thorough than a single game.
- While they have a dataset of human guides for each level, it's extremely small, with only a few hundred levels total.
- The evaluation is generally sparse and uncompelling. They have no quantitative metrics at all, their qualitative evaluation is not impressive, and their evaluation is based on an extremely small number of examples anyway.
- The paper also doesn't even seem complete. There are only 3 sections, a bit over 5 pages, etc.

I have a number of other concerns, but these are enough to make this paper seem like a clear reject.

**Summary Of The Paper:**

This paper suggests using deep learning to learn expert decision-making from game strategy guides. They focus on the task of Angry Birds, for which they have natural language descriptions of what actions to take to pass a given level. Given a visual representation of an Angry Birds screenshot, they train an LSTM to model the natural language guides. They claim that their model generalizes well.

**Summary Of The Review:**

The evaluation of this paper is extremely limited and uncompelling.

---

> ### Author Response · Authors · 2021-11-22
> **response to reviewer 2**
>
> We would like to thank the reviewer’s response. We did have difficulty finding proper datasets for our goal and decided to focus on Angry Bird game (AB) for two reasons. First, it demands players to plan multiple actions ahead. Second, its environment contains abstract objects, allowing DL models to detect objects easily and focus on predicting human players’ strategies. Using AB, we focused on testing if neural networks can learn human players' strategies from screenshots using MLPs and ABIC.
>
> However, we do understand the reviewer’s concern about the limited amount and scope of the dataset. We are currently looking for other datasets to extend our study.

---

### Official Review · Reviewer_7AXP · 2021-11-03

**Correctness:** 1
**Technical Novelty And Significance:** 2
**Empirical Novelty And Significance:** 2
**Recommendation:** 3
**Confidence:** 3

**Main Review:**

The paper is clearly written and the experimental setup is clearly provided. The topic is interesting, but I'm not sure if the contribution is novel enough or if the model/experiment achieve the given goal.

The model or the task/data is not novel. The combination can be novel. However, predicting the optimal strategy description from the screenshot does not seem to serve as a planning task (or a task that involves a sequence of actions). The difficulty of this task can come from an unknown outcome from a pre-defined action, e.g. the physical interaction is unknown and needs to be learned from experience. Thus, the next action cannot be pre-defined without the outcome of the previous action. The experiment in this paper completely remove this layer of complexity by only trying to produce the 3SG.

The paper lacks baseline model comparison.

Instead of mapping from a problem to a solution (2.1), the model might just be doing object detection. There are only a small set of possible objects in the screenshot and in the strategy description. The training setup in 2.2.1 seems to encourage object detection than a problem to solution mapping. It ignores the temporal relationship of actions of different stages. It's not surprising that the results show that EAB can only describe the first action, especially given the authors argument the screenshot with the first bird.

**Summary Of The Paper:**

The paper proposes to use DL to predict the optimal strategy (in words) for angry birds levels. It applies a image captioning model to predict the optimal strategy description using the screenshot of the setup of different levels.

**Summary Of The Review:**

The topic of the paper is interesting, but overall the paper lacks novelty and contribution, and the experiments are not carefully designed and done, and with no comparison to other models

---

> ### Author Response · Authors · 2021-11-22
> **response to reviewer 1**
>
> First of all, we would like to thank the reviewer’s suggestions. In this study, we asked if DL models can mimic humans’ high-level planning. As our daily tasks involve sequential actions, we tested if DL models (agents) can predict a sequence of actions using the Angry Bird game (AB) as a testbed. As pointed out by the reviewer, playing AB is difficult, since agents need to understand physical interactions between various objects and evaluate outcomes of actions. However, we did not focus on teaching agents to play AB properly; we focused on testing agents’ ability to learn to organize sequential actions by ‘reading’ human players’ strategies. Specifically, inspired by the fact that we can create a plan to clear AB stages before taking any action, we trained EAB with screenshots only.
>
> We understand the reviewer’s concern that EAB’s success can be boiled down to object detection. We would like to point out that our experiments with synthetic images could address this concern (at least, partially). When detecting objects becomes easier, EAB’s answers improved substantially, suggesting that EAB can distribute resources between object detection and problem solving (i.e., mappings between screenshots and strategies). Finally, we agree with the reviewer that we need to extend the scope of our experiments to address our goal more properly and are currently planning on future works.

---

### Decision · Program_Chairs · 2022-01-20

**Decision:**

Reject

**Comment:**

This paper trains a neural network to predict expert strategies (described in natural language) in the game of Angry Birds. While the reviewers agreed this was potentially interesting, there was also a consensus that the scope of the paper was too narrow, that the writing was imprecise, and that the evaluations too few and too qualitative. I agree the paper does not seem thorough enough for ICLR, and recommend rejection.